# Incidence and prevalence of hypertension among HIV-TB co-infected participants accessing treatment: A five-year prospective cohort analysis

Halima Dawood[1,2]* , Nonhlanhla Yende-Zuma[1,3] , Upasna Singh[1] , Mikaila C. Moodley[1] , Jenine Ramruthan[1] , Kogieleum Naidoo[1,3] *

1 Centre for the AIDS Programme of Research in South Africa (CAPRISA), Durban, South Africa,
2 Department of Internal Medicine, Infectious Diseases, Greys Hospital, Pietermaritzburg, South Africa,
3 MRC-CAPRISA HIV-TB Pathogenesis and Treatment Research Unit, Doris Duke Medical Research Institute, University of KwaZulu-Natal, Durban, South Africa

◉ These authors contributed equally to this work.
* Halima.Dawood@caprisa.org (HD); Kogie.Naidoo@caprisa.org (KN)

**Data Availability Statement:** The data dictionary, and datasets used and/or analysed during the current study are available. The full final clinical

## Abstract

### Introduction

Hypertension is a leading risk factor for cardiovascular disease among people living with human immunodeficiency virus (PLWH). This study determined incidence and prevalence of hypertension among PLWH receiving antiretroviral therapy (ART).

### Method

We prospectively followed-up 642 HIV and tuberculosis (TB) co-infected study participants from 2005–2013. We defined hypertension as two consecutive elevated systolic and/or diastolic blood pressure measurements above 139/89 mmHg or current use of antihypertensive therapy.

### Results

Of 507 participants analyzed, 53% were women. Median [interquartile range (IQR)] age, body mass index (BMI), and CD4 count was 34 (28.0–40.0) years, 22.7 (20.5–25.4) kg/m$^2$, and 145 (69.0–252.0) cells/mm$^3$, respectively. Incidence [95% confidence interval (CI)] of both systolic and diastolic hypertension overall, in men, and in women over 40 years was 1.9 (1.4–2.6), 5.9 (3.6–9.6), and 5.0 (2.7–9.3) per 100 person-years (PY), respectively. Risk of developing hypertension was higher in men [(adjusted hazard ratio (aHR) 12.04, 95% CI: 4.35–33.32)] and women over 40 years (aHR 8.19, 95% CI 2.96–22.64), and in men below 40 years (aHR 2.79, 95% CI 0.95–8.23).

data set will be available to through a request lodged on the CAPRISA website (http://www.caprisa.org/Pages/CAPRISA%20Studies). The written request wil be assessed by the CAPRISA Scientific Review Committee within 30 business days of submission.

**Funding:** The TRUTH study was supported by the Howard Hughes Medical Institute, Grant Number 55007065, as well as the Centers for Disease Control and Prevention (CDC) Cooperative Agreement Number UY2G/PS001350-02. Its contents are solely the responsibility of the authors and do not necessarily represent the official views of either the Howard Hughes Medical Institute or the Centers for Disease Control and Prevention (CDC). Patient care was supported by the KwaZulu-Natal Department of Health and the U.S. President's Emergency Plan for AIDS Relief (PEPFAR). The funding sources listed here did not have any role in the study design, data collection and analysis, decision to publish, or preparation of the data in this manuscript, nor was any payment received by these or other funding sources for this manuscript.

**Competing interests:** The authors declare no competing interests.

**Abbreviations:** AIDS, acquired immunodeficiency syndrome; ART, ART Antiretroviral therapy; BMI, Body Mass Index; CAPRISA, Centre for the AIDS Programme of Research in South Africa; CI, confidence interval; D: A: D, Data Collection on Adverse events of Anti-HIV Drugs; ELISA, enzyme-linked immunosorbent assay; GEE, generalized estimating equation; HIV/AIDS, human immunodeficiency virus; HR, Hazard ratio; IQR, interquartile range; JNC, Joint National Committee; OR, odds ratio; PLWHA, People living with HIV/AIDS; PY, Person-year; SAPIT, Starting antiretrovirals at three Points in TB treatment; SAS, Statistical Analysis System; SD, Standard deviation; TB, Tuberculosis; TRUTH, TB Recurrence upon Treatment with HAART; WHO, World Health Organization.

## Conclusion

Higher incidence rates of hypertension among older men and women accessing ART highlight opportunities to expand current integrated HIV-TB care models, to include cardiovascular disease risk screening and care to prevent premature death.

## Introduction

Cardiovascular disease is the leading cause of death worldwide, and the second leading cause of death after Human Immunodeficiency Virus (HIV)/ acquired immunodeficiency syndrome (AIDS) in South Africa [1–3]. While widespread use of antiretroviral therapy (ART) among people living with HIV/AIDS (PLWHA) contribute to increased life expectancy and reduced AIDS related mortality, these patients are at heightened risk for several non-AIDS complications commonly associated with ageing [4, 5]. Hypertension remains the leading risk factor for mortality from cardiovascular disease with an estimated one and a half-to- two-fold greater risk among PLWHA despite effective ART, posing a hidden threat to global HIV control [6]. The success of South Africa's large ART program is likely to be offset by major clinical challenges arising from management of long-term non-communicable diseases including cardiac, renal, metabolic neurological, pulmonary, and oncological complications of ART.

Published data from the anti-HIV drugs (D: A: D) study and Swiss HIV Cohort Study indicate a high prevalence of hypertension over time in PLWHA on ART [7, 8]. While the epidemiology of hypertension among PLWHA within high-income settings is well defined, there is currently no accurate estimation of the prevalence, and incidence rates within low-income disease endemic regions [5, 9, 10]. In 2013, it was estimated that the overall prevalence of hypertension within a South African population was 30.4% in those 15 years and older [11]. In a similar population where hypertension was defined as blood pressure greater than 140/90mmHg, increasing age-associated prevalence of hypertension was observed among PLWHA within age groups of those younger than 34 (31.4%), 35–44 (40.6%), 45–54 (58.1%), and older than 55 years (53.7%) [12].

Distinct traditional risk factors for hypertension in HIV uninfected populations are well described in literature [13]. However, data on demographic factors, genetic predisposition, lifestyle, ageing population, and pre-existing co-morbidities among PLWHA within an ART era do not comprehensively explain the exacerbated risk of hypertension in low-income settings. Additionally, studies report HIV-dependent non-traditional risk factors for hypertension in PLWHA including duration on ART, endothelial dysfunction and other ART related adverse effects [14].

Comprehensive estimates of incidence and prevalence of hypertension among PLWHA on ART within a disease endemic South African setting is limited. We report incidence and prevalence of hypertension among ART accessing PLWHA with previous TB, followed up over five years.

## Methods

### Study design and participants

The Starting antiretrovirals at three Points in HIV and TB treatment trials (SAPIT) enrolled HIV-TB co-infected patients between 2005 to 2008 in Durban, KwaZulu-Natal, South Africa. Participants in the SAPIT randomized controlled trial were subsequently followed up in the

CAPRISA 005 TB Recurrence upon Treatment with HAART (TRuTH) study, a prospective cohort study assessing TB recurrence among patients initiated on ART between 2009 and 2013. This study prospectively collected clinical and demographic data over five years of 642 HIV-TB co-infected participants across these two cohorts. Details of the cohort, procedures and the primary outcomes of the studies have been described previously [4, 15]. We conducted a secondary analysis of 507 adult (18 years and older) HIV-TB co-infected patients initiating ART. This study was approved by the Biomedical Research Ethics Committee of the University of KwaZulu-Natal (SAPIT ref no:e107/05 and TRUTH ref no : BF 051/09 TRUTH). The study was conducted in accordance with the relevant guidelines and regulations. All participants were informed of the potential benefits and risks. Written Informed consent was obtained from all study participants.

## Study procedures

Antiretroviral therapy initiation date was taken as baseline. All patients received standard of care as per national department of health guidelines at the time, duration and type of ART was recorded [16]. Confirmation of HIV-infection was based on two successive rapid HIV Enzyme linked immunosorbent assay (ELISA) tests. Patient information including demographic information, detailed medical history and a full evaluation of the current clinical condition was undertaken at screening, study enrolment and initially monthly for the first six months post-ART initiation, then every 2–3 months, unless clinically indicated. Routine safety laboratory tests and CD4+ counts (FACS flow cytometer: Becton Dickinson, Franklin Lakes NJ, USA) and viral loads (Roche Cobas Amplicor HIV-1 Monitor v1.5) were performed at baseline and six-monthly. These details, along with blood pressure measurements and medication prescribed, were captured in real-time on case report forms (CRFs) and submitted to a central CAPRISA electronic data management system (DFdiscover, DF/Net Research, Inc). On every visit systolic and diastolic blood pressure was recorded with the participant seated and the elbow at heart level using a digital sphygmomanometer, weight and height were measured using standard methods. All vital checks followed standard operating protocol. Study participants were given a unique study number making them unidentifiable individually. Data is not available on a public platform and was anonymized before analysis.

## Definition of hypertension

Based on the Joint National Committee (JNC 8) and South African standard treatment guidelines we defined hypertension as systolic measurements ≥140 mmHg mmHg and diastolic measurements ≥90 mmHg on two consecutive visits or current use of antihypertensive therapy [17, 18].

## Statistical analysis

The longitudinal follow up included all patients in the clinical trial and subsequent cohort study on ART between June 2005 and July 2008 (SAPIT) and from 2009–2013 for TRUTH study respectively. Demographic and clinical variables were summarized using medians with interquartile range (IQR), mean with standard deviation (SD) and percentages.

To account for variation in blood pressure readings, we used logistic regression model using generalized estimating equations (GEE) to identify predictors associated with elevated blood pressure measurements over time. The multivariable model was adjusted for age, gender, smoking, and alcohol drinking measured at ART initiation and also body mass index (BMI) which was treated as a time-varying co-variate. Alcohol use and cigarette smoking were not used as a time varying variable as we did not collect this information with the monthly

blood pressure measurements. We considered hypertension diagnosed within three months of ART initiation as prevalent hypertension. Additionally, we conducted an analysis using proportional hazards model to identify the predictors of incident hypertension post ART initiation. This analyses excluded participants diagnosed with prevalent hypertension. We accounted for the same variables as listed above. The incidence rates per 100 person-years (PY) were calculated using a Poisson model with person-years as an offset. Kaplan-Meier curve was used to explore the timing of incident hypertension post ART initiation. Statistical analysis was done using SAS (version 9.4.; SAS Institute Inc., Cary, NC, USA).

## Results

### Baseline clinical and demographic characteristics at ART initiation

Among 507 HIV-infected participants initiated on ART 52.5% were women, median (IQR) age [min-max], Body mass index (BMI), CD4 count, urea, and creatinine levels was 34 years (28.0–40.0)[19–72]; 22.7 (20.5–25.4) kg/m2, 145 (69.0–252.0) cells/mm3, 3.4 (2.7–4.3) mmol/l, and 72.0 (63.0–82.0) umol/l, respectively. (Table 1). Average viral load was 5.0 log copies/ml [standard deviation (SD) 0.9] and haemoglobin was 11.3 g/dl (SD 2.0) (Table 1). Cigarette smoking and alcohol intake were equally prevalent at approximately 15%. (Table 1). Median time on ART among study participants was 5.2 (IQR 2.2–6.1) years.

### Incidence and prevalence of hypertension

A total of 21 840 systolic and diastolic blood pressure measurements were recorded from 507 participants, n = 1322 (6.1%) episodes of elevated systolic blood pressure only, n = 2001 (9.2%) episodes of elevated diastolic blood pressure only, and n = 779 (3.6%) episodes of concurrently raised systolic and diastolic blood pressure. Incident rates [95% confidence interval (CI)] of hypertension for systolic, diastolic, and both systolic and diastolic blood pressure was 3.1 (95% CI 2.4–3.9), 5.3 (95% CI 4.4–6.4), and 1.9 (95% CI 1.4–2.6) per 100 person years (PY), respectively (Table 2). Study participants diagnosed with hypertension had a median time on ART (months) of 45 (IQR 13.0–59.0). Incidence of hypertension stratified by age and gender was 5.9 (95% CI 3.6–9.6), and 5.0 (95% CI 2.7–9.3) per 100 PY in men and women 40 years and older, respectively (Table 2). Overall, we observed 42 hypertension events: 6 among 214 women, and 10 among 164 men younger than 40 years of age, and 10 among 50 women, and 16 among 71 men 40 years and older. Compared to men and women younger than 40 years of age, incidence of hypertension was four-fold higher in men 40 years and older at 5.9 per 100 PY, and eight-fold higher in women 40 years and older at 5.0 per 100 PY (Table 2 and Fig 1). Overall, elevated systolic blood pressure only, elevated diastolic blood pressure only, and concurrent elevated systolic and diastolic blood pressure occurred in 75 (14.8%), 124 (24.5%), and 50 (9.9%) of participants respectively. Of which 46 participants were on antihypertensive medication by the end of the follow-up period. In the first three months of ART initiation 11 participants had elevated systolic blood pressure only and 18 participants had elevated diastolic blood pressure only. We observed prevalent hypertension in 1.6% (n = 8) of study participants, five men and two women over 40 years, and one man under 40 years.

### Risk factors associated with hypertension

Compared to women below 40 years men and women 40 years and older on ART had a significantly higher risk of incident systolic hypertension (adjusted hazard ratio (aHR) 9.21, 95% CI 4.11–20.60); (aHR 6.55, 95% CI 2.96–14.49), diastolic hypertension (aHR 4.60, 95% CI 2.54–8.35); (aHR 4.41, 95% CI 2.45–7.93), and both systolic and diastolic hypertension (aHR 12.04,

**Table 1. Clinical and demographic characteristics at ART initiation.**

| Variable | Overall (N = 507) |
|---|---|
| **Demographic factors** | |
| Age (years), median (**IQR) | 34.0 (28.0–40.0) |
| Females, n (%) | 266 (52.5%) |
| *Employment status, n (%)* | |
| Scholar | 9 (1.8%) |
| Employed | 302 (59.6%) |
| Unemployed | 196 (38.7%) |
| **Metabolic factors** | |
| *Body mass index group, n (%)* | |
| <18.5 kg/m$^2$ | 34 (6.7%) |
| 18.5–24.99 kg/m$^2$ | 328 (64.7%) |
| 25–29.99 kg/m$^2$ | 97 (19.1%) |
| ≥ 30 kg/m$^2$ | 48 (9.5%) |
| Urea (mmol/l), median(IQR) | 3.4 (2.7–4.3) |
| Creatinine (umol/l), median(IQR) | 72.0 (63.0–82.0) |
| Smoking at ART initiation | 76 (15.1%) |
| **HIV Related factors** | |
| *Regimen, n (%)[b]* | |
| EFV/3TC/AZT | 2 (0.4%) |
| EFV/3TC/DDI | 37 (7.3%) |
| EFV/3TC/DDIEC | 460 (90.9%) |
| EFV/3TC/TDF | 4 (0.8%) |
| EFV/COMBIVIR | 1 (0.2%) |
| NVP/3TC/DDIEC | 2 (0.4%) |
| CD4 count (cells/mm$^3$), median (IQR) | 145.0 (69.0–252.0) |
| Viral load (log$_{10}$ copies/ml), mean (*SD) | 5.0 (0.9) |
| Haemoglobin (g/dl), mean (SD) | 11.3 (2.0) |
| Male | 12.1 (2.0) |
| Female | 10.5 (1.7) |

**IQR Interquartile range

* SD: standard deviation,135 participants were not initiated on ART and therefore were only included in the demographic's factors

95% CI 4.35–33.32); (aHR 8.19, 95% CI 2.96–22.64), respectively (p-value < .001) (Table 3). A similar trend was observed for men below 40 years.

## Risk factors associated with elevated blood pressure measurements

Men 40 year and older, women 40 year and older, and men younger than 40 years had higher odds of elevated systolic and diastolic blood pressure over time as (adjusted odds ratio (aOR) 9.36, 95% CI: 4.98–17.59), = (aOR 4.43, 95% CI 2.10–9.33),and (aOR 2.65, 95% CI 1.33–5.29). Moreover, higher BMI (aOR 1.62, 95% CI 1.39–1.88), was associated with an increased odds of elevated systolic and diastolic blood pressure over time (Table 4). Variables associated with elevated systolic only and diastolic only are shown in Table 4.

**Table 2. Incidence rate (/100py) of hypertension amongst patients initiating ART overall, by age and gender.**

| Variable | | †N | ††Person years | number of HPT events | Incidence rate/100py |
|---|---|---|---|---|---|
| **Systolic only** | | | | | |
| | Overall | 496 | 2097.17 | 64 | 3.1 (2.4–3.9) |
| | Women below 40 years | 213 | 987.74 | 11 | 1.1 (0.6–2.0) |
| | Men 40 years and above | 69 | 243.59 | 21 | 8.6 (5.6–13.2) |
| | Women 40 years and above | 50 | 181.44 | 14 | 7.7 (4.6–13.0) |
| | Men below 40 years | 164 | 684.4 | 18 | 2.6 (1.7–4.2) |
| **Diastolic only** | | | | | |
| | Overall | 489 | 1991.36 | 106 | 5.3 (4.4–6.4) |
| | Women below 40 years | 214 | 962.9 | 27 | 2.8 (1.9–4.1) |
| | Men 40 years and above | 70 | 240.2 | 25 | 10.4 (7.0–15.4) |
| | Women 40 years and above | 47 | 162 | 20 | 12.3 (8.0–19.1)) |
| | Men below 40 years | 158 | 626.26 | 34 | 5.4 (3.9–7.6) |
| **Both systolic and diastolic** | | | | | |
| | Overall | 499 | 2168.67 | 42 | 1.9 (1.4–2.6) |
| | Women below 40 years | 214 | 995.91 | 6 | 0.6 (0.3–1.3) |
| | Men 40 years and above | 71 | 273.02 | 16 | 5.9 (3.6–9.6) |
| | Women 40 years and above | 50 | 200.29 | 10 | 5.0 (2.7–9.3) |
| | Men below 40 years | 164 | 699.45 | 10 | 1.4 (0.8–2.7) |

†N = total number of patients at risk

†† PY:person-years, hypertensiondefined as systolic measurements ≥140 mmHg and diastolic measurements ≥90 mmHg on two consecutive visits or curre nt use of antihypertensive therapy

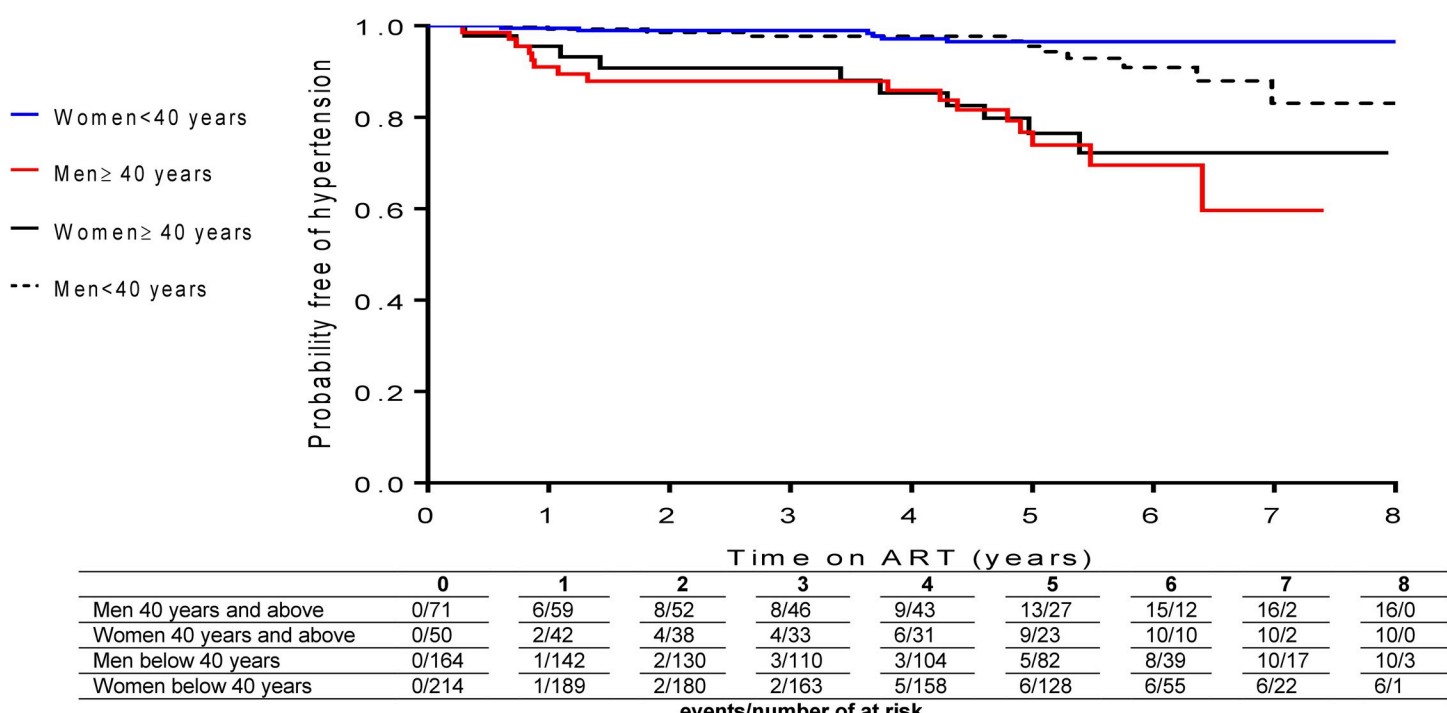

**Fig 1. Kaplan-Meier survival estimates of incident hypertension in PLWHA accessing ART.**

**Table 3. Univariable and multivariable analyses of factors associated with incident hypertension.**

| Variable | Systolic ***BP | | | | Diastolic ***BP | | | | Both systolic and diastolic ***BP | | | |
|---|---|---|---|---|---|---|---|---|---|---|---|---|
| | Univariable | | Multivariable | | Univariable | | Multivariable | | Univariable | | Multivariable | |
| | §HR (95% #CI) | p-value | §HR (95% #CI) | p-value | §HR (95% #CI) | p-value | §HR (95% #CI) | p-value | §HR (95% #CI) | p-value | §HR (95% #CI) | p-value |
| Women below 40 years | reference | | | | | | | | | | | |
| Men 40 years and above | 8.11 (3.90–16.86) | < .001 | 9.21 (4.11–20.60) | < .0001 | 4.14 (2.39–7.15) | < .001 | 4.60 (2.54–8.35) | < .0001 | 10.36 (4.04–26.56) | < .001 | 12.04 (4.35–33.32) | < .0001 |
| Women 40 years and above | 6.94 (3.15–15.29) | < .001 | 6.55 (2.96–14.49) | < .0001 | 4.68 (2.62–8.36) | < .001 | 4.41 (2.45–7.93) | < .0001 | 8.57 (3.11–23.61) | < .001 | 8.19 (2.96–22.64) | < .0001 |
| Men below 40 years | 2.41 (1.14–5.10) | 0.022 | 2.73 (1.21–6.16) | 0.015 | 1.97 (1.19–3.27) | 0.008 | 2.21 (1.26–3.86) | 0.005 | 2.40 (0.87–6.60) | 0.09 | 2.79 (0.95–8.23) | 0.063 |
| Smoking at ART initiation | 1.37 (0.70–2.69) | 0.365 | 0.73 (0.33–1.64) | 0.450 | 1.40 (0.82–2.38) | 0.217 | 1.04 (0.55–1.97) | 0.909 | 1.50 (0.67–3.39) | 0.328 | 1.06 (0.39–2.84) | 0.916 |
| Alcohol drinking at ART initiation | 2.02 (1.15–3.57) | 0.015 | 1.75 (0.88–3.47) | 0.111 | 1.44 (0.89–2.34) | 0.142 | 1.12 (0.62–2.01) | 0.715 | 1.61 (0.77–3.36) | 0.207 | 1.17 (0.47–2.89) | 0.732 |
| BMI per 5 kg/m² increase‡, | 1.00 (0.80–1.25) | 0.980 | 1.20 (0.94–1.54) | 0.144 | 1.02 (0.86–1.21) | 0.810 | 1.14 (0.94–1.38) | 0.184 | 1.00 (0.76–1.31) | 0.987 | 1.20 (0.88–1.62) | 0.255 |

‡,time-varying

§HR: hazard ratio; blood pressure (***BP); confidence-interval (#CI), The following participants were included: N = 496 for Systolic BP, N = 489 for Diastolic BP, N = 499 for Both systolic and diastolic.

## Discussion

Compared to published data from South Africa and other developed settings we observed substantially lower overall incidence rates of hypertension among PLWHA followed up over 5 years [19–22]. Incident hypertension defined in published literature was a single elevated systolic and diastolic blood pressure measurement. This likely contributed to the higher incidence rates reported in these studies [8, 19, 20]. Furthermore, other notable differences in the population of PLWHA with hypertension in comparator studies include older age, higher socio-

**Table 4. Analyses of risk factors associated with elevated blood pressure over time.**

| Variable | Systolic ***BP | | | | Diastolic ***BP | | | | Both systolic and diastolic ***BP | | | |
|---|---|---|---|---|---|---|---|---|---|---|---|---|
| | Univariable | | Multivariable | | Univariable | | Multivariable | | Univariable | | Multivariable | |
| | ¶OR (95% #CI) | p-value | ¶OR (95% #CI) | p-value | ¶OR (95% CI) | p-value | ¶OR (95% #CI) | p-value | ¶OR (95% #CI) | p-value | ¶OR (95% #CI) | p-value |
| Women below 40 years | reference | | | | | | | | | | | |
| Men 40 years and above | 6.39 (3.94–10.35) | < .001 | 10.06 (5.67–17.87) | < .001 | 3.14 (2.23–4.43) | < .001 | 5.01 (3.42–7.34) | < .001 | 6.79 (3.74–12.33) | < .001 | 9.36 (4.98–17.59) | < .001 |
| Women 40 years and above | 5.47 (3.13–9.56) | < .001 | 4.37 (2.29–8.35) | < .001 | 3.00 (2.03–4.43) | < .001 | 2.39 (1.45–3.93) | < .001 | 5.91 (3.02–11.54) | < .001 | 4.43 (2.10–9.33) | < .001 |
| Men below 40 years | 1.96 (1.20–3.18) | 0.007 | 2.65 (1.50–4.71) | < .001 | 1.59 (1.16–2.17) | 0.004 | 2.33 (1.57–3.46) | < .001 | 2.22 (1.17–4.22) | 0.015 | 2.65 (1.33–5.29) | 0.006 |
| Smoking at ART initiation | 1.32 (0.80–2.18) | 0.28 | 0.81 (0.38–1.72) | 0.577 | 1.30 (0.91–1.87) | 0.147 | 0.92 (0.54–1.57) | 0.764 | 1.51 (0.85–2.66) | 0.158 | 0.85 (0.33–2.14) | 0.724 |
| Alcohol drinking at ART initiation | 1.44 (0.91–2.26) | 0.119 | 1.66 (0.72–3.84) | 0.236 | 1.52 (1.10–2.10) | 0.01 | 1.56 (0.92–2.67) | 0.102 | 1.62 (0.99–2.65) | 0.055 | 1.69 (0.64–4.44) | 0.287 |
| BMI per 5 kg/m² increase‡ | 1.53 (1.39–1.69) | < .001 | 1.71 (1.48–1.97) | < .001 | 1.58 (1.43–1.74) | < .001 | 1.74 (1.55–1.97) | < .001 | 1.46 (1.30–1.63) | < .001 | 1.62 (1.39–1.88) | < .001 |

‡time-varying; odds ratio (¶OR); confidence-interval (#CI)

economic status, longer duration on ART, a predominance of Caucasian patients, and high baseline Framingham score, which likely contributed to the higher hypertension incidence rates observed. Nonetheless, our reported incidence rates of hypertension stratified by age is consistent with other studies conducted among PLWHA [8, 19, 21]. Despite hypertension being among the leading cause of death worldwide and hypertension prevalence rates being widely reported, data on age standardized incidence rates of hypertension in the general population remain scarce. Compared to our findings of prevalent hypertension among 1.6% of PLWHA aged 19 to 72, global age-standardized prevalence of hypertension in 2019 was 32% in women and 34% in men aged 30–79 years [23]. Interestingly, irrespective of gender, prevalence of hypertension from low and middle-income countries was markedly higher compared to high-income countries [23, 24]. This highlights the inequality gap influencing social determinants of health, lack of universal healthcare coverage, and low physician-to-patient ratios consequently contributing to established risk factors for hypertension including unhealthy diet with high salt intake, physical inactivity, tobacco, alcohol use, and obesity [25].

As reported elsewhere, we observed time on ART to be an associated risk factor for hypertension among both men and women, especially those 40 years and older [8, 18, 19]. Diagnosis of hypertension was made upon program entry only in those 40 years and older, likely due to previously undiagnosed hypertension. While all age groups demonstrate no notable increased hypertension incidence rates in the first four years of follow-up, risk of hypertension increased substantially in men and women 40 years and older and increased moderately in men younger than 40 years. Conversely, women younger than 40 years showed no increased risk of hypertension during follow-up. Compared to ART naïve patients, significantly higher rates of both systolic and diastolic blood pressure levels among ART-exposed patients and ART experienced PLWHA (OR 1.68, 95% CI 1.35–2.10), was reported in a systematic review and meta-analysis [26], highlighting the association between cumulative ART exposure and hypertension. The data in our study highlight that PLWHA, irrespective of age, would benefit from regular screening for hypertension. Additionally, findings from this study support integration of chronic disease screening and management into ART programs.

There was a significant association between increasing age in both male and females 40 years and older and elevated systolic and diastolic ($p < 0.001$) blood pressure. Males younger than 40 years were also at significant risk of incident systolic hypertension [HR: 2.7, (95% CI: 1.2–6.2)], $p = 0.015$, and incident diastolic hypertension [HR: 2.2, (95% CI: 1.3–3.9)], $p = 0.005$, compared to female counterparts in the same age group. Cigarette smoking and alcohol intake were not associated with incident hypertension. A quarter of all participants had a BMI > 25 kg/m$^2$, however BMI was not associated with incident hypertension but with elevated isolated systolic or diastolic blood pressures. Multifaceted risk factors for hypertension in the general population are similar to those in PLWHA [10, 21]. As with general population studies, we found a significant association between increasing age in participants 40 years and older and elevated systolic and diastolic blood pressure. Contrary to known risk factors for hypertension in the general population cigarette smoking, alcohol intake and a BMI > 25 kg/m$^2$ were not associated with incident hypertension in this cohort.

Published data demonstrates that PLWHA experience HIV related risk factors that contribute to hypertension [8]. We identified similar risk factors including low pre-ART CD4 count, increasing ART duration, and exposure to ART drugs such as stavudine, didanosine, zidovudine, nevirapine and drugs belonging to the protease inhibitors class. It is noteworthy that many of the ART drugs taken by our study participants although no longer in use, are known to confer heightened risk for hypertension. The current World Health Organization (WHO) standard of care ART regimen for resource limited settings includes dolutegravir known to be associated with substantive weight gain, increased BMI, and body fat distribution changes, all

of which are significant contributors to hypertension [27, 28]. A recent Zambian report demonstrated that the use of a dolutegravir based regimen is a significant risk factor for hypertension [29].

Co-morbid HIV and hypertension may remain asymptomatic for long periods of time, hence early screening and diagnosis of hypertension and other non-communicable diseases will assist in timeous management and control of these conditions [30, 31]. Therefore, policies supporting integration of screening and treatment for non-communicable diseases in HIV treatment programs are urgently required [32]. Emerging evidence from donor-funded projects suggest that HIV and non-communicable disease service integration highlight advantages that boost co-management including; reduction in duplication of records, fragmentation of services, co-delivered services for improved retention in care and treatment adherence to prevalent co-morbidities [33]. In the current South African Ideal Clinic model, deficiencies within the care cascade are addressed to provide stronger service integration of chronic disease stream for HIV and hypertension management [34]. However, further task-shifting and decentralized care is warranted to inform policy.

Our findings suggest that annual blood pressure screening will detect two cases of hypertension for 100 persons on long term ART. We do, however, acknowledge limitations of our study by the observational study design and failure to measure associations between individual drug exposure and raised blood pressure as most patients where on a single standardized regimen. White coat hypertension was not investigated or excluded. This report did not include specific lifestyle and metabolic risk factors including physical activity, diet, high salt or low potassium intake, and regular alcohol consumption. The absence of a HIV-negative comparator population limited our ability to analyze age-standardized incidence of hypertension in our study. These may contribute to confounding and bias, limiting the generalizability of our results. Blood pressure measurements in this study were done with the patient's elbow at heart level and this reading may vary depending on the position of the arm. However, the strength of the study was the ability to follow-up blood pressure measurements over time and hence measurement variability was reduced.

Future research to understand temporal trends in hypertension incidence among maturing ART cohorts is urgently required. Evidence-based data providing insights into the complete cardiovascular risk profile of young PLWHA with co-morbid hypertension, within disease endemic areas is required to help anticipate future health needs.

## Conclusion

We found high incidence rates of hypertension stratified by age among men and women 40 years and older receiving a longer duration of ART. Implementation science research assessing evidence-based strategies to reduce the risk of hypertension in low-income disease endemic regions is warranted. Maturing HIV programs require effective mechanisms of identifying patients with cardiovascular disease risk factors, and systems for triage of high risk PLWHA to prevent premature morbidity and mortality. The aging HIV population on ART globally will benefit from expanding the HIV treatment guidelines to include an integrated approach that includes targeted screening and close clinical observation for chronic non-communicable diseases.

## Acknowledgments

We acknowledge the contribution of the SAPIT and TRuTH Study participants and the effort of the entire treatment team at CAPRISA eThekwini clinical site who worked on the parent study.

## Author Contributions

**Conceptualization:** Halima Dawood, Kogieleum Naidoo.

**Data curation:** Nonhlanhla Yende-Zuma.

**Formal analysis:** Halima Dawood, Nonhlanhla Yende-Zuma, Kogieleum Naidoo.

**Funding acquisition:** Halima Dawood.

**Methodology:** Halima Dawood, Nonhlanhla Yende-Zuma, Kogieleum Naidoo.

**Writing – original draft:** Halima Dawood, Jenine Ramruthan, Kogieleum Naidoo.

**Writing – review & editing:** Halima Dawood, Nonhlanhla Yende-Zuma, Upasna Singh, Mikaila C. Moodley, Jenine Ramruthan, Kogieleum Naidoo.

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
