## [Decision Letter · Decision Letter 0]

22 Aug 2023

PONE-D-23-16476Incidence and prevalence of hypertension among HIV-infected participants accessing treatment: A five-year prospective cohort analysisPLOS ONE

Dear Dr. Dawood,

Thank you for submitting your manuscript to PLOS ONE. After careful consideration, we feel that it has merit but does not fully meet PLOS ONE’s publication criteria as it currently stands. Therefore, we invite you to submit a revised version of the manuscript that addresses the points raised during the review process.

We look forward to receiving your revised manuscript.

Kind regards,

Joel Msafiri Francis, MD, MS, PhD

Academic Editor

PLOS ONE

“Research reported in this publication was partly supported by the South African Medical Research Council (SAMRC).”

“We acknowledge the contribution of the SAPIT and TRuTH Study participants and the effort of the entire treatment team at CAPRISA eThekwini clinical site who worked on the parent study. Research reported in this publication was, supported by the South African Medical Research Council (SAMRC). The TRUTH study was supported by the Howard Hughes Medical Institute, Grant Number 55007065, as well as the Centers for Disease Control and Prevention (CDC) Cooperative Agreement Number UY2G/PS001350-02. Its contents are solely the responsibility of the authors and do not necessarily represent the official views of either the Howard Hughes Medical Institute or the Centers for Disease Control and Prevention (CDC). Patient care was supported by the KwaZulu-Natal Department of Health and the U.S. President’s Emergency Plan for AIDS Relief (PEPFAR). The funding sources listed here did not have any role in the analysis or preparation of the data in this manuscript, nor was any payment received by these or other funding sources for this manuscript.”

“Research reported in this publication was partly supported by the South African Medical Research Council (SAMRC).”

Reviewers' comments:

Reviewer's Responses to Questions

**Comments to the Author**

1. Is the manuscript technically sound, and do the data support the conclusions?

Reviewer #1: Yes

Reviewer #2: Yes

2. Has the statistical analysis been performed appropriately and rigorously? 

Reviewer #1: Yes

Reviewer #2: I Don't Know

3. Have the authors made all data underlying the findings in their manuscript fully available?

Reviewer #1: Yes

Reviewer #2: Yes

4. Is the manuscript presented in an intelligible fashion and written in standard English?

Reviewer #1: Yes

Reviewer #2: Yes

5. Review Comments to the Author

Reviewer #1: This is a well written manuscript.

However, since a lot has already been published on this research topic, I am wondering why this analysis was conducted / manuscript was written?, what knowledge gaps you aimed to fill by this analysis?/what does this manuscript add to what we already know about HIV and Hypertension?

Also, this was HIV - TB cohort, were there any impact of TB treatment in HIV on incidence of Hypertension?

Reviewer #2: General remarks

Congratulations to the authors for this piece of work highlighting the growing burden of cardiovascular risk conditions in PLWHA co-infected with TB. Findings from this study would help to guide appropriate intervention for early detection and treatment of hypertension in PLWHA.

General comments

Participants for this study were HIV/TB coinfected. I would suggest this to be emphasised in the title.

The authors used antiretroviral treatment and antiretroviral therapy interchangeably. I suggest using antiretroviral therapy throughout the abstract and the main text

All abbreviations in the abstract and main text should be defined on first appearance in the text.

Specific comments

Methods

Line 137: “The start of the ART initiation in the SAPiT study was taken as baseline” – Do the author mean “ART initiation date was taken as baseline”.

Lines 149 – 151: Procedures for blood pressure measurements need to be explained in detail.

Specify guidelines did you use for BP measurement. Specify the model of automated BP sphygmomanometer, weight scale and stadiometer

Lines 156-157: Specify guidelines used to define hypertension

Lines 160 - 162: Definition for hypertension does not reflect the cited guideline (JNC8) i.e., “systolic measurements above 139 mmHg and/or diastolic measurements above 89 mmHg on two

162 consecutive visits or current use of antihypertensive therapy”

Lines 162 – 163: The sentence “Patients already on medication for hypertension were considered to be hypertensive” is a repetition of the information already included in the definition for hypertension. Consider deleting this sentence of rephrase the definition.

Data analysis

The authors chose to define hypertension occurring during the first 3 months post enrolment as prevalent hypertension. What was the basis to choose 3 months’ time-frame in the definition?

The authors considered BMI as a time varying covariate in the multivariable logistic regression model but did not allow BMI as a time varying co-variate in the Poisson regression model. Would be helpful to the reader to understand why BMI was not treated as time varying co-variate in the Poisson model.

Results

642 participants were followed-up but the results are available for 507 participants. What was the reason for not including 135 participants? How do the 135 participants not included in the analysis compare with the remaining 507 in terms of baseline characteristics?

Line 278 – 282: comparison of adjusted Hazard Ratios (aHRs) should mention the reference group indicated in table 3

Table 2

- HPT should be defined on the table footer

- Variables with missing values should be indicated

Table 3

- How many participants were included in the regression analysis?

- All abbreviations (i.e., BP, CI) should be defined on the table footer. The symbol § should appear on in all HR abbreviations

Table 4

- How many participants were included in the regression analysis?

- All abbreviations (i.e., BP, CI etc.) needs to be defined on the table footer

6. PLOS authors have the option to publish the peer review history of their article (what does this mean?). If published, this will include your full peer review and any attached files.

Reviewer #1: No

Reviewer #2: No

---

## [Author Response · Author response to Decision Letter 0]

16 Nov 2023

General remarks:

1. However, since a lot has already been published on this research topic, I am wondering why this analysis was conducted / manuscript was written?, what knowledge gaps you aimed to fill by this analysis?/what does this manuscript add to what we already know about HIV and Hypertension? Also, this was HIV - TB cohort, were there any impact of TB treatment in HIV on incidence of Hypertension?

Response: Cardiovascular disease is the second leading cause of death after HIVAIDS in South Africa, and hypertension remains the leading risk factor for mortality from cardiovascular disease among PLWHA, posing a hidden threat to global HIV control. We describe high incidence rates of hypertension stratified by age among men and women 40 years and older, receiving a longer duration of ART. This study highlights that maturing HIV programs require effective mechanisms of identifying patients with cardiovascular disease risk factors, and systems for triage of high risk PLWHA to prevent premature morbidity and mortality.

2. Congratulations to the authors for this piece of work highlighting the growing burden of cardiovascular risk conditions in PLWHA co-infected with TB. Findings from this study would help to guide appropriate intervention for early detection and treatment of hypertension in PLWHA.

Response: We thank the reviewers for their helpful comments and questions.

General comments:

3. Participants for this study were HIV/TB coinfected. I would suggest this to be emphasised in the title. 

Response: We thank reviewers this has been corrected. 

4. The authors used antiretroviral treatment and antiretroviral therapy interchangeably. I suggest using antiretroviral therapy throughout the abstract and the main text

Response: We thank reviewers this has been corrected.

5. All abbreviations in the abstract and main text should be defined on first appearance in the text. 

Response: We thank reviewers this has been corrected.

Specific comments:

 Methods

6. Line 137: “The start of the ART initiation in the SAPiT study was taken as baseline” – Do the author mean “ART initiation date was taken as baseline”.

Response: line 139“The start of the ART initiation in the SAPiT study was taken as baseline.” The authors refer to ART initiation date as baseline. This sentence as been amended and now reads “Antiretroviral therapy initiation date was taken as baseline.”

7. Lines 149 – 151: Procedures for blood pressure measurements need to be explained in detail. Specify guidelines did you use for BP measurement. Specify the model of automated BP sphygmomanometer, weight scale and stadiometer.

Response: Standard operating protocols (SOP) are provided to guide the measurement of all vital signs including BP, weight, and height. All clinic staff are trained on the practices and procedures involved in ascertaining vital checks, and this service is rendered to all patients seeking primary healthcare services at the clinic. This study was conducted between 2009-2013, therefore the exact model name and manufacturing details of all apparatus used can not be retrieved currently. 

Lines 151-156 has been amended and now reads “On every visit systolic and diastolic blood pressure was recorded with the participant seated and the elbow at heart level using a digital sphygmomanometer, weight and height were measured using standard methods. All vital checks followed standard operating protocol. Study participants were given a unique study number making them unidentifiable individually. Data is not available on a public platform and was anonymized before analysis.”

Measurement of BP was ascertained using a digital blood pressure monitoring sphygmomanometer.

• The patient is seated in an upright position with their arm stretched out straight and adequately exposed to the middle half of the upper arm for placement of the BP cuff. 

• The bladder of the cuff is placed centrally over the brachial artery that was palpated on the ante-cubital fossa approximately 1/3 of the way from the medial epicondyle. The middle of the rubber inflatable bag is placed over the inner side of the upper arm and the cuff is wrapped evenly around the arm and secured by tucking in the end or by strapping it onto the Velcro surface. 

• The digital machine is then switched on to allow the BP reading process. Once the reading processing is complete the BP reading will appear on the screen which will include the pulse rate of the participants. 

• This information is thereafter recorded on the participants vital log. 

• These readings were recorded in the vital log sheet. All readings of systolic BP >140 or diastolic BP >90 was re-taken, and the second recording was entered into the vitals log and captured onto the e-database. 

Blood pressure was measured after 1 and 3 minutes of standing at first consultation in the elderly, diabetics and in participants where orthostatic hypotension is common. All abnormal readings were repeated with the initial and repeat reading recorded onto the Vitals logs. The study nurse was trained immediately inform the study clinician of any relevant abnormalities detected. 

Weight

• A digital scale is used to obtain the patient weight in kilograms. 

• The patient was asked to remove shoes and any other heavy clothing as well other heavy items. 

• The patient was then instructed to stand on the base of the scale whilst maintaining an erect posture. 

• The scale start button is then switched on, and the weight reading is recorded on the vitals log sheet. 

Height

• A digital scale with calibrations in both centimetres and feet is used. 

• The patient was asked to remove shoes and to stand on the base of the scale whilst maintaining a relaxed and erect posture. 

• The adjustable arm on the scale was moved above the patient’s head and rests just slightly on the patient’s head. 

• A reading was taken where the arrow on the scale points to the specific calibration. This reading was recorded in centimetres only. 

• The recording is entered on the vitals log sheet. 

8. Lines 156-157: Specify guidelines used to define hypertension.

Response: We thank the reviewers. Lines 157-159 has been removed “Definition of Elevated Blood Pressure-Elevated blood pressure was calculated as systolic measurements above 139 mmHg and/or diastolic measurements above 89 mmHg on each visit.”

Lines 161-167 has been amended and now reads-

“Definition of Hypertension 

Based on the Joint National Committee (JNC 8) and South African standard treatment guidelines we defined hypertension as systolic measurements ≥140 mmHg mmHg and diastolic measurements ≥90 mmHg on two consecutive visits or current use of antihypertensive therapy [17,18].”

Reference 18 has been updated on the reference list to-“The National Department of Health, South Africa: Essential Drugs Programme. Primary Healthcare Standard Treatment Guideline and Essential Medicine List. 2008. South African National Department of Health” 

9. Lines 160 - 162: Definition for hypertension does not reflect the cited guideline (JNC8) i.e., “systolic measurements above 139 mmHg and/or diastolic measurements above 89 mmHg on two consecutive visits or current use of antihypertensive therapy”

Response: Based on JNC8 and the South African standard treatment guidelines(2008) we considered participants with systolic blood pressure (SBP) ≥140 mmHg or diastolic blood pressure (DBP) ≥90 mmHg as hypertensive. This has been amended, lines161-167 now reads

 ”Definition of Hypertension 

Based on the Joint National Committee (JNC 8) and South African standard treatment guidelines we defined hypertension as systolic measurements ≥140 mmHg above 139 mmHg and diastolic measurements ≥90 mmHg above 89 mmHg on two consecutive visits or current use of antihypertensive therapy [17,18].”

10. Lines 162 – 163: The sentence “Patients already on medication for hypertension were considered to be hypertensive” is a repetition of the information already included in the definition for hypertension. Consider deleting this sentence of rephrase the definition.

Response: We apologise for the error. this sentence has been removed. 

 Data analysis

11. The authors chose to define hypertension occurring during the first 3 months post enrolment as prevalent hypertension. What was the basis to choose 3 months’ time-frame in the definition?

Response: In clinical practice a diagnosis of hypertension is made with 2 readings at least on two different visits with 2-3 months. The rationale for our “time-frame” was to exclude those with undiagnosed hypertension at study entry and to allow for observations to coincide with monthly scheduled visits.

12. The authors considered BMI as a time varying covariate in the multivariable logistic regression model but did not allow BMI as a time varying co-variate in the Poisson regression model. Would be helpful to the reader to understand why BMI was not treated as time varying co-variate in the Poisson model.

Response: We apologise for the confusion. We used time-varying BMI in all our models and that is why there is a footnote in Table 3 showing that BMI was a time-varying variable. However, the statistical analyses section was not updated. Lines 183-187 has been amended and now reads :

“These analyses excluded participants diagnosed with prevalent hypertension. We accounted for the same variables as listed above. The incidence rates per 100 person-years (PY) were calculated using a Poisson model with person-years as an offset.”

 Results

13. 642 participants were followed-up but the results are available for 507 participants. What was the reason for not including 135 participants? How do the 135 participants not included in the analysis compare with the remaining 507 in terms of baseline characteristics?

Response: The 135 participants were not initiated on ART and therefore excluded from the analyses. Given that they were not initiated on ART, we can only compare them on three demographic factors listed in Table 1 (i.e. age, gender and employment status). 

This as been added to the footnote of table 1 

“135 participants were not initiated on ART and therefore were only included in the demographics factors”

14. Line 278 – 282: comparison of adjusted Hazard Ratios (aHRs) should mention the reference group indicated in table 3.

Response: We thank reviewers. This has been added to line 278. 

15. Table 2-HPT should be defined on the table footer, Variables with missing values should be indicated.

Response: We thank reviewers. Hypertension has been defined in the table footer. As variables were measured when patients attended follow-up the only missingness was a patient at the time of a follow-up visit. 

16. Table 3-How many participants were included in the regression analysis?

All abbreviations (i.e., BP, CI) should be defined on the table footer. The symbol § should appear on in all HR abbreviations.

Response: All abbreviations have been defined in the table footer and symbol § appears on all HR abbreviations. Sample sizes for proportional hazards regression models varies because this analysis excludes prevalent cases for each outcome variable. 

The following participants were included: N=496 for Systolic BP, N=489 for Diastolic BP, N=499 for Both systolic and diastolic. This has been added to the footnote on table 3.

17. Table 4-How many participants were included in the regression analysis? All abbreviations (i.e., BP, CI etc.) needs to be defined on the table footer.

Response: All abbreviations have been defined in the table footer. A total of 507 were included in all the logistic regression models with all their time-varying BP measurements. 

18. The following additional changes have been made:

• addition to table 1 footnote- “135 participants were not initiated on ART and therefore were only included in analysis of demographics factors”

• addition to table 3 and table 4 footnote-“blood pressure (***BP), confidence-interval (#CI)

• addition to table 3 and 4-symbols “*** and # “ were added to abbreviations BP and CI respectively. 

• Lines 395-398 has been amended and now reads “These may contribute to confounding and bias, limiting the generalizability of our results. Blood pressure measurements in this study were done with the patient’s elbow at heart level and this reading may vary depending on the position of the arm.”

• Due to corrections- line numbers and reference numbers have changed.

• Funding related information has been removed from the acknowledgement section

“Funding 

Research reported in this publication was, supported by the South African Medical Research Council (SAMRC). The TRUTH study was supported by the Howard Hughes Medical Institute, Grant Number 55007065, as well as the Centers for Disease Control and Prevention (CDC) Cooperative Agreement Number UY2G/PS001350-02. Its contents are solely the responsibility of the authors and do not necessarily represent the official views of either the Howard Hughes Medical Institute or the Centers for Disease Control and Prevention (CDC). Patient care was supported by the KwaZulu-Natal Department of Health and the U.S. President’s Emergency Plan for AIDS Relief (PEPFAR). The funding sources listed here did not have any role in the study design, data collection and analysis, decision to publish, or preparation of the data in this manuscript, nor was any payment received by these or other funding sources for this manuscript.” This has been added to the online submission and the cover letter.

---

## [Decision Letter · Decision Letter 1]

2 Jan 2024

Incidence and prevalence of hypertension among HIV-TB co-infected participants accessing treatment: A five-year prospective cohort analysis

PONE-D-23-16476R1

Dear Dr. Dawood,

We’re pleased to inform you that your manuscript has been judged scientifically suitable for publication and will be formally accepted for publication once it meets all outstanding technical requirements.

Kind regards,

Joel Msafiri Francis, MD, MS, PhD

Academic Editor

PLOS ONE

Additional Editor Comments (optional):

Reviewers' comments:

Reviewer's Responses to Questions

**Comments to the Author**

1. If the authors have adequately addressed your comments raised in a previous round of review and you feel that this manuscript is now acceptable for publication, you may indicate that here to bypass the “Comments to the Author” section, enter your conflict of interest statement in the “Confidential to Editor” section, and submit your "Accept" recommendation.

Reviewer #1: All comments have been addressed

2. Is the manuscript technically sound, and do the data support the conclusions?

Reviewer #1: Yes

3. Has the statistical analysis been performed appropriately and rigorously? 

Reviewer #1: Yes

4. Have the authors made all data underlying the findings in their manuscript fully available?

Reviewer #1: Yes

5. Is the manuscript presented in an intelligible fashion and written in standard English?

Reviewer #1: Yes

6. Review Comments to the Author

Reviewer #1: The authors have adequately attended to all the raised comments. Kindly consider to accept the article for publication

7. PLOS authors have the option to publish the peer review history of their article (what does this mean?). If published, this will include your full peer review and any attached files.

Reviewer #1: No

---

## [Editor Report · Acceptance letter]

18 Jan 2024

PONE-D-23-16476R1 

PLOS ONE

Dear Dr. Dawood, 

I'm pleased to inform you that your manuscript has been deemed suitable for publication in PLOS ONE. Congratulations! Your manuscript is now being handed over to our production team.

Kind regards, 

on behalf of

Dr. Joel Msafiri Francis 

Academic Editor

PLOS ONE